# An In Vitro Evaluation of the Capacity of Local Tanzanian Crude Clay and Ash-Based Materials in Binding Aflatoxins in Solution

**DOI:** 10.3390/toxins10120510

**Published:** 2018-12-03

**Authors:** Emmanuel M. Ayo, Athanasia Matemu, Germana H. Laswai, Martin E. Kimanya

**Affiliations:** 1School of Life Sciences and Bio-engineering, The Nelson Mandela African Institution of Science and Technology, P.O. Box 447, Arusha 23000, Tanzania; ayoe@nm-aist.ac.tz or eayo@irdp.ac.tz (E.M.A.); athanasia.matemu@nm-aist.ac.tz (A.M.); 2Institute of Rural Development Planning, P.O. Box 138, Dodoma 41000, Tanzania; 3Department of Animal, Aquaculture and Range Sciences, College of Agriculture, Sokoine University of Agriculture, P.O. Box 3004, Morogoro 67000, Tanzania; laswaig@sua.ac.tz

**Keywords:** clays, ashes aflatoxins, binding capacity, in vitro and contaminated feeds

## Abstract

Aflatoxins in feeds cause great health hazards to animals, and thus eventually to humans as well. The potential of clays from Arusha (AC), Kilimanjaro (KC), the Coast (CC), and Morogoro (MC), as well as volcanic ash (VA) and rice husk ash (RA), were evaluated for their capacity to adsorb aflatoxins B_1_ (AFB_1_), B_2_ (AFB_2_), G_1_ (AFG_1_), and G_2_ (AFG_2_) relative to a commercial binder Mycobind^®^ (R) using in vitro technique. On average, CC, VA, KC, MC, AC, RA, and R adsorbed 39.9%, 51.3%, 61.5%, 62.0%, 72.6%, 84.7%, and 98.1% of the total aflatoxins from solution, respectively. The capacity of AC and RA was statistically (*p* < 0.05) better in binding aflatoxins next to R. The adsorption capacity seemed to follow the trend of the cation exchange capacity (CEC) of these materials. The CEC (meq/100 g) of CC, MC, KC, VA, AC, RA, and R were 7.0, 15.4, 18.8, 25.4, 27.2, 27.2, and 38.9, respectively. On average 96.3%, 42.7%, 80.8%, and 32.1% of AFB_1_, AFB_2_, AFG_1_, and AFG_2_ were adsorbed, respectively. The binding capacity of the clays and ashes relative to Mycobind^®^ was about 100% for AC and RA, 50% for KC, MC, and VA, and 33.3% for CC. The AC and RA seem to be promising resources in binding aflatoxins in solution.

## 1. Introduction

Aflatoxins are natural toxins that are produced in foods and feeds, primarily by certain species of fungi, specifically *Aspergillus flavus* and *Aspergillus parasiticus*, when conditions are favorable for fungal growth and subsequent toxin formation. Aflatoxins exist in four forms of health, agricultural importance, and economic importance, namely aflatoxin B_1_ (AFB_1_), B_2_ (AFB_2_), G_1_ (AFG_1_), and G_2_ (AFG_2_) [1,2,3]. The most toxic and abundant aflatoxin is AFB_1_ [4]. Almost all feed resources contain certain levels of naturally occurring aflatoxins, and any level of dietary aflatoxins consumed poses a certain health risk [5]. Studies show that aflatoxins in feeds depress the growth and production performance of animals [6,7]. When animals are fed naturally aflatoxin-contaminated feeds, the toxins (mostly AFB_1_) are secreted in cow milk or retained in hen eggs as aflatoxin M1 (AFM1) [8,9,10].

Therefore, it is imperative to prevent and reduce hazards of aflatoxin contamination of feeds for the protection and promotion of human and animal health. Techniques used to reduce aflatoxin contamination in feeds include physical methods such as thermal inactivation and irradiation; chemical techniques such as the treatment of the feeds with acidic or alkaline solutions, ozone treatment, and ammoniation; and biological techniques such as detoxification by microbial agents [11,12]. These techniques are mostly applied in the animal industry, and are reported to have some limitations including cost implications, the requirement of some complicated facilities, the reduction of dietary palatability and nutritional values, and the created danger of unsafe chemical residual [13]. Techniques involving the use of toxin binders (also called adsorbents or sequesters) have also been employed, owing to their economic feasibility, applicability, and nutritional safety.

Many types of crude or refined materials, including clays, cellulose products, yeast cell wall products, and activated charcoal products are envisaged to have the ability to sequester or bind aflatoxins [14,15,16]. The potential binding capacity of these materials are known to vary depending on their nature and source [17]. However, the generic relationship between the binding effectiveness and absorbent properties of aflatoxins, including their elemental and mineralogical content, cation exchange capacity (CEC), and pH levels, is not yet clearly established [17,18]. The binding potential of some materials, particularly clays, seems to be the function of their chemical composition, such as Ca^+^ and K^+^ ions present in the framework configured by silicon, aluminum, and iron oxide. Studies show that aluminosilicates have a wide variation of these elements (Table 1). In South American countries, for example, ashes such as soda ash and wood ash have been used in some food processes such as in nixtamalization for corn tenderization, where dietary aflatoxins load is also reduced, owing to the breakage of the aflatoxin lactone ring by the ash alkalinity [19]. In Tanzania, farmers use some imported binders such as Mycobind^®^ that seem to be useful in terms of protecting livestock from aflatoxin exposure (personal communication with Farmers Centre Limited). However, the imported binders are expensive. Therefore, the high cost of importing these clay-based materials can be avoided if local resources and sources with similar potential are identified. Experience in animal husbandry in Tanzania shows that there is a number of clay and ash-based materials that are directly eaten by human or added to feeds and foods for various purposes [20].

The clay-based materials are sold for geophagial purposes, and are mostly demanded by some groups of women, especially pregnant ones. Ashes have been used in the traditional cookery of some local foods such as corn recipes, and in feeds as ration improvers or appetizers.

We hypothesize that these materials could provide an aflatoxin-binding capacity that is equivalent to the imported product. Arbitrarily, we selected clays obtained in the regions of Arusha, Kilimanjaro, Morogoro, and the Coast, as well as volcanic ash and rice husk ash from the Arusha region. The potential of these materials in binding toxins has been speculated from the instinct of them being among the eating soils of animals, birds, and humans, which shield them from the toxic effects of some ingested natural toxins [27,28].

Therefore, the objective of the study was to evaluate the chemical composition and the capacities of the above-mentioned materials in binding aflatoxins.

## 2. Results

### 2.1. Chemical Composition of the Test Binding Material and the Reference Binder

The major minerals contained in the test binding materials (TBM) and reference binder (R) are shown in Table 2. Muscovite mineral was observed in clays from Arusha (AC) and Kilimanjaro (KC), kaolinite was observed in the clays from Coast (CC) and Morogoro (MC), leucite was observed in the clay from Morogoro (MC), microline and ephicite were observed in volcanic ash (VA), albite and terranovite were observed in rice husk ash (RA), while metanatrolite and phlogopite were observed in reference binder (R).

Table 3 shows the elemental (oxide) composition of the test binding material (TBM) and reference binder (R). All of the samples of TBM and R contained aluminum and silicon elements as the backbone of the minerals. Other important elements that were observed as parts of the chemical formula of the prominent minerals in the TBM and R were iron in AC, VA, and R; calcium in VA, RA, and R, and potassium in all of the materials except CC and RA. The VA and R had minerals containing all of the main elements: aluminum, silicon, iron, calcium, and potassium. 

RA showed the lowest content of aluminum oxide (alumina) of 0.5%; all of the other TBM had content above that of R at 5.1%. Percent silicon oxide (silica) contents in CC and RA were above that of R, while the other TBM had contents from 22–32.8%, which was a level lower than that of R (49%). The VA and RA had percent contents of potassium oxide that were a little bit higher than that of R. The VA had calcium oxide content that was a bit higher than that of R, while the rest of the TBM had percent contents below that of R. The AC and RA had the highest and the lowest contents of iron oxide, respectively. Except for RA and CC, which had a lower percent of iron oxide contents, AC, KC, MC, and VA had values above that of R.

The values of CEC for the TBM are also shown in Table 3. The values of CEC for the TBM ranged from 7 meq/100 g for CC to 27.2 meq/100 g for RA. All of the TBM had lower values of cation exchange capacity (CEC) compared to that of Mycobind^®^ (38.9 meq/100 g).

### 2.2. Capacity of the Binders to Bind Aflatoxin

The percent of aflatoxin-binding capacity of the test binding material (TBM) and reference binder (R) are shown in Table 4 (across the columns). The percent binding capacity of the TBM ranged from a minimum value of 40 for CC to a maximum value of 85 for RA, which were relative to 98 for R. The mean proportions of aflatoxins as adsorbed by the TBM and R are also shown on Table 4 (across the rows). The proportions of aflatoxins that were adsorbed were relatively high for AFB_1_ and AFG_1_, and low for AFG_2_ and AFB_2_.

The relationship of elemental (oxide) concentration in the test binding material (TBM) and reference binder (R), and their cation exchange capacity (CEC) is shown in Table 5. The relationship as correlation coefficients was positive and relatively higher with CaO (0.63), K_2_O (0.59), and Fe_2_O_3_ (0.11), and negative with SiO_2_ (−0.06) and Al_2_O_3_ (−0.86). Similarly, the relationship between the elemental (oxide) concentration in the binding materials, TBM and R (BM), and their capacity to bind total aflatoxins, was positive and relatively higher with K_2_O (0.51), CaO (0.34), SiO_2_ (0.21), Fe_2_O_3_ (0) and Al_2_O (−0.88). The relationship between the CEC of the BM and their capacity to bind total aflatoxins was relatively high, with a correlation coefficient of 0.90 (Table 5).

The equivalence of Mycobind^®^ in binding the total aflatoxins relative to the TBM is shown in Table 6. The binding capacity of Mycobind^®^ was one fold that of the clay from Arusha (AC) and rice husk ash (RA), twice that of the clays from Kilimanjaro (KC) and Morogoro (MC) and volcanic ash (VA), and thrice that of the clay from the Coast (CC). The capacity of the Mycobind^®^ to bind AFB_1_ was one fold that of all of the test binding materials

## 3. Discussion

Generally, the chemical composition of the binding material (BM) that was observed in this study was similar to the composition of aluminosilicate-based materials reported from various other studies. For instance, the alumina content of the materials was within the range reported in other studies of 0.45–21.7% [21] and 13.2% (Massinga et al., 2010) as cited by Anjos et al. [23], except for clay from the Coast (CC), which contained a higher level of alumina, at about 33%. Except for rice husk ash (RA), which showed a much higher percent content of silica, the other BM had content similar to the reported values for clay materials, ranging from 1.1–69.0% (mean of 59.6%) [21] and 44.3–71.0% (mean of 55.3%) [20]. Similar to Mycobind^®^, volcanic ash (VA) and RA had potassium oxide content above the previously reported range of 0.1–3.3% [19] and 0.1–2.6% [21] and 0.1% (Massinga et al., 2010), cited by Anjos et al. [23] for high aflatoxin binding. The 0.01% potassium oxide content of clay from Kilimanjaro (KC) was below the reported levels. The calcium oxide content in all of the BM were found to be within the previously reported range of 0.1–31.4% [21,22,23] for clay materials Except for the CC and RA, the rest of the BM showed iron oxide content above the previously reported range of 0.2–14.8% for binders [19,21] according to Massinga et al. (2010), as cited by Anjos et al. [22]. From the comparative composition of BM, it seems that the materials do not differ from other materials of aluminosilicate nature, including those proved to bind aflatoxins.

Aluminosilicate-based materials have been reported to have cation exchange capacity (CEC) (meq/100 g) values ranging from 10 (kaolinite mineral) to 100 (illite and smectite minerals) and medium values are found around the value of 25 [29]. Except for the CC, which showed a low value of 7 meq/100 g, the rest of the TBM had CEC values within the documented range, as were observed from 15.4 meq/100 g for clay from Morogoro (MC) to 38.9 meq/100 g for Mycobind^®^.

Among the evaluated binding materials, RA and the clay from Arusha (AC) had binding capacity almost equal to that of the Mycobind^®^, the reference binder, particularly in binding AFB_1_ and AFG_1_, which are the most toxic aflatoxins. It is possible that the excellent binding power of these materials was due to their relatively high CEC values. The CEC values of both RA and AC were 27.2 meq/100 g of the materials, and are equivalent to the CEC value for the reference material. The high CEC values of many binding materials have been reported to promote their capacity to bind aflatoxins [17]. The relatively high values of calcium (Ca^2+^) and potassium (K^+^) content in the aluminosilicate minerals of the evaluated materials seemed to promote the CEC values of the materials. Studies have shown that concentrations of Ca^2+^ and K^+^ ions make a great contribution to CEC levels in aluminosilicate materials [29]. The presence of silicon (Si^4+)^, aluminum (Al^3+^), and iron (Fe^3+^) seemed to have low or negative influence on the CEC values of the BM. According to Brow and Lemom [30], values of CEC increase with decreasing acidity, and vice versa. The ions Si^4+^, Al^3+^, and Fe^3+^ promote the acidity of materials in solution, unlike Ca^2+^ and K^+^ [31]; hence, they negatively influence the CEC values of the BM, and subsequently their capacity to bind aflatoxins in solution. Disregarding other factors such as the structural effect of a material, it is probable that materials such as CC showed a low capacity for aflatoxin binding partly due to their higher concentration of Al^3+^ and Si^4+^, and partly due to their relatively higher content of a kaolinite type of mineral, which has a low CEC [30]. Furthermore, KC and MC could not bind aflatoxins efficiently, which was possibly due to their relatively higher concentration of Al^3+^ and Fe^3+^.

X-ray diffraction (X-RD) analysis showed the presence of prominent mineral components that can influence aflatoxin binding in the BM. The results showed that similar to the Mycobind^®^, RA and AC contained major minerals such as andranite/melanite, terranovite, and albite; all of these contain calcium and phlogopite, as well as muscovite, which contains potassium. It is possible that these components rendered RA and AC relatively superior to others in binding aflatoxins. In aflatoxin-binding ions, Ca^2+^ in particular synchronously bonds to two aflatoxin carbonyls, and at the same time binds to the four oxygen atoms of the Si–O ring on the clay binder surface [32]. However, AC had low Ca^2+^ and K^+^ cations, yet its CEC value was relatively high enough to favor high aflatoxin-binding capacity. Seemingly, the way that active cations such as calcium and potassium are incorporated in different structures of the BM, and their associations with other structural elements, may affect the adsorptive potential of the BM.

The results for the aflatoxin-binding capacity of the BM concurred with the results of other previous related in vitro studies in which the binding capacity levels of clay-based binders such as bentonites (about 90%) have been reported [33,34]. The Mycobind^®^ that was employed as a reference in this study missed the manufacturer’s information displaying its capacity to bind aflatoxins. However, in our analysis, we found that it could bind about 98% of the total aflatoxins that were subjected to it. A similar product that was evaluated in Kenya, Agrolite-Mycobind^®^, showed an aflatoxin-binding capacity of 95% [35]. Regarding the minimum experimental set-up standards as suggested by Whitlow [36]—although ours were slightly higher—the binding capacity of 98% that was observed for the Mycobind^®^ in this study matched closely to the 95% value reported for the Agrolite-Mycobind^®^.

The binding capacity ratio of Mycobind^®^ to the test binding materials (TBM) observed in this study conversably implied that AC and RA bind 100%, KC, MC and VA bind 50%, and CC binds 33.3% of the total aflatoxins in solution. All of the TBM had the capacity to bind AFB_1_ equivalent to the Mycobind^®^. This indicates that although they exist in varying levels, the locally available crude materials (clay and ash-based resources) have the potential to adsorb aflatoxins in solution media, and can possibly reduce the aflatoxin contamination of feeds. 

The AFB_1_ and AFG_1_ were highly adsorbed into the TBM as compared to AFB_2_ and AFG_2_. This is most likely because compared to AFB_2_ and AFG_2_, the AFB_1_ and AFG1 have a higher polarity of the β-dicarbonyl group, which is a key functional group of the aflatoxins [14]. With respect to the polarity, AFB_1_ was rendered the most adsorbed by the TBM, followed by AFG_1_. This is advantageous, since the toxicity of the aflatoxins tends to follow this order of reactivity, which was also obeyed by our results in this study. The aflatoxin-binding capacity of the evaluated materials (especially RA and AC) can be confirmed through an in vivo test where the dietary and animal’s gastrointestinal tract (GIT) factors are automatically accommodated. However, since exported binders are costly to farmers in low-income countries, the material can be occasionally used in feeds to reduce the hazardous effects of aflatoxins on animals. In addition, traditionally, farmers have been using an array of such materials for various intentions, including uses in animal feeds. It has been observed that wild animals and birds are less affected by many natural toxins, which probably includes aflatoxins, owing to their geophagial instincts [27,28]. Essentially, these animals and birds fetch and eat clayey soil, which renders them safe from the inherent food toxins [37]. Through a number of studies, including those reported by Phillips et al. [16], Denli et al. [38], and Kaoud [39], clay soil has been proved to detoxify aflatoxins. Harnessing this natural phenomenon may be economically helpful to poor farmers as one of the strategies for lowering aflatoxin menace, which is difficult to avoid in feeds.

## 4. Conclusions

The test materials that we evaluated in the study had varying capacity levels of binding aflatoxins in solution. The crude materials AC (Arusha clay) and RA (rice husk ash) have the relatively higher potential to bind aflatoxins equivalent to the commercial product Mycobind^®^, which was employed in the study for reference purposes. Since these cheap materials have been traditionally used for various purposes in animals, occasionally they could be utilized to minimize the exposure of aflatoxin load to animals through contaminated feeds. Further studies are recommended to test the binding capacity of these materials in refined form, and when used in combinations of two or more, using both on in vitro and in vivo trials.

## 5. Materials and Methods

### 5.1. Materials

#### 5.1.1. Test Binding Materials and Their Sources

Six crude test binding materials (TBM) were evaluated against a commercial binder, Mycobind^®^, as the reference material (R). Of these TBM, four were clays, designated AC, KC, CC, and MC, and two were ash-based materials, volcanic ash (VA) and rice husk ash (RA). The nature, source, and ethno-utilization of the TBM are shown in Table 7. Samples of AC, KC, CC, and MC were obtained by taking several aliquots of a material from various parts of the source or sampling lot, making representative samples of about five kilograms of each TBM. The samples were taken to the laboratory at the Geological Survey of Tanzania for cleaning, grinding, sieving, and homogenization; then, they were packed in zip bags for subsequent chemical analyses and evaluation for adsorption capacity.

Five kilograms of VA was purchased from the market, and the site of production was followed to ascertain its originality; then, it was handled similarly to the clay samples in the laboratory. A representative sample of rice husks was taken from rice millers and incinerated in the laboratory furnace at the Nelson Mandela African Institution of Science and Technology (NM-AIST) at a temperature of 550 °C for four hours, producing about five kilograms of rice husk ash.

#### 5.1.2. The Reference Binder

For comparison of the binding capacity of the crude clays and ashes, a commercial mycotoxin detoxifier named Mycobind^®^ (Evonik Industries AG) was purchased from Farmers Centre Limited in Dar es Salaam, Tanzania and employed in the study.

#### 5.1.3. Aflatoxin Solution

The stock solution of aflatoxins (Romer Labs, Inc., Washington, MO, USA) was donated by the Tanzania Food and Drugs Authority (TFDA).

### 5.2. Chemical Analysis of the Test Materials

Samples of the binding materials (BM) were further homogenized, ground, and sieved through a one-millimeter sieve for the subsequent analyses of mineralogical composition, elemental content, and cation exchange capacity (CEC).

#### 5.2.1. Mineralogical Composition

Samples of the BM were analyzed for their mineralogical composition using non-destructive techniques that employed an X-ray diffraction (X-RD) analyzer (BTX SN 231, Olympus Corporation, Tokyo, Japan) or a self-calibrated XRD analyzer, depending on temperature. The samples were analyzed at a temperature of −45 °C. About 15 mg of finely ground sample was sieved through a 150-µm sieve and loaded in the vibrating sample holder of the X-RD analyzer for scanning. The results were X-RD spectrum patterns that were received on a screen of a computer connected to the analyzer, showing peaks corresponding to each specific mineral present in the sample.

#### 5.2.2. Elemental (Oxide) Composition

The oxides in the BM were quantified by Minipal-4, which was a high-performance bench top energy dispersive X-ray fluorescence spectrometer (PANalytical MINIPAL-4, EDXRF Spectrometer, Almelo, The Netherlands). The sample was ground into a fine powder; then, about 50 g of it was scanned by the spectrometer for metallic oxide composition at an energy dispersion of 30 keV. The percent composition of the metallic oxides in each sample was recorded.

#### 5.2.3. Determination of Cation Exchange Capacity

The cation exchange capacity (CEC) was determined by wet analysis employing the ammonium replacement method (Buchner funnels vacuum flasks), as explained by Brady and Weil [35] and involving the leaching of exchangeable cations in the binding materials (BM) with ammonium acetate salt solution. The excess salt was removed by ethanol and followed by potassium chloride to leach NH_4_^+^, which initially replaced other various cations of the BM. The amount of NH4^+^ that was released and washed into a beaker beneath Buchner funnels was determined using the Kjeldahl distillation method [40], and the CEC (meg/100 g) of BM was computed using Equation (1):CEC = (mg L^−1^ of NH_4_-N in leachate) × (0.25/14) × (100/sample weight (g)) mg L^−1^ NH_4_-N(1)

### 5.3. Experimental Design and Treatments

#### 5.3.1. Experimental Design

The BM that was engaged to bind aflatoxins formed seven treatments of the in vitro experiment. Each of the treatments was replicated into three units (test tubes).

#### 5.3.2. Preparation of the Experimental Solutions

The experiment was based on a buffer solution with or without a TBM/Mycobind^®^ and spiked or non-spiked with aflatoxin solution.

a. Buffer Solution

The buffer solution was prepared from potassium chloride, potassium dihydrogen phosphate, anhydrous disodium hydrogen phosphate, and sodium chloride in distilled water.

b. Diluted Aflatoxin Solution

The standard solution of combined aflatoxins AFB_1_, AFB_2_, AFG_1_, and AFG_2_ (250 ng/mL) in acetonitrile was diluted to 20 ng/mL using distilled water in an amber flask.

c. Solutions of BM and Controls

The test solutions contained components as shown below, which are also summarized in Table 8.
(i)Spiked TBM/Mycobind^®^: suspension of 0.25% of BM in the buffer solution spiked with 5 mL of diluted solution of aflatoxins.(ii)Non-spiked BM: a control for each binding material containing suspension of 0.25% of BM in the buffer solution.(iii)Positive control: the buffer solution spiked with 5 mL of diluted solution of aflatoxins; and(iv)Negative control: the buffer solution only.

For each solution, three replications were taken in separate test tubes as experimental units.

#### 5.3.3. Procedure for In Vitro Experiment

The in vitro procedure was adopted from Kong et al. [36], simulating the gastrointestinal pH condition of pigs and representing monogastric animals, which are more prone to aflatoxicosis. A sample of TBM/Mycobind^®^ was prepared by weighing 0.025 g into 10 mL of phosphate buffer solution (0.1 M, pH 6.0) making a suspension of 0.25%. An aliquot of 2.5-mL suspension was pipetted into 25-mL centrifuge-tube; then, 5 mL of the diluted aflatoxin solution was added. Parallel with the BM test treatments, their respective negative controls (non-spiked with the diluted aflatoxin solution) were run. General positive and negative controls were included to eliminate the probable error effects such as due to aflatoxin impurities in the measuring/analysis system hardware and reagents. The positive control contained 2.5 mL of phosphate buffer, and five mL of the diluted aflatoxin solution, while the negative control contained five mL of the phosphate buffer solution only. Each solution sample was replicated thrice, and the pH in each centrifuge tube was adjusted to 2.0 by adding 1 M of HCl to simulate the pH in the stomach of pigs.

#### 5.3.4. Incubation of the Solution Samples

All of the samples were incubated at 39 °C in a shaking water bath for two hours; then, one milliliter of phosphate buffer (0.2 M, pH 6.8) was added to each tube. To simulate the conditions in the small intestine of pigs, the pH in all of the tubes was raised to 6.8 by adding 1 M of NaOH, followed by a second phase of incubation at 39 °C for four hours. After incubation, the mixture was centrifuged, and the supernatant was obtained for an analysis of the residual (unbound) aflatoxins B1, B2, G1, and G2 using high-performance liquid chromatography (HPLC).

### 5.4. Determination of Unbound Aflatoxins in the Buffer Solution

Briefly, the pH of the clear supernatant was adjusted to 7.4 using 0.1 M of NaOH. Unbound aflatoxin in the supernatant was determined by the procedure suggested by Diaz and Smith [11], where the clear supernatant was analyzed for residual (unbound) aflatoxin without additional cleanup. The analysis employed a fluorescence detector connected to an HPLC (Shimadzu Corp, Kyoto, Japan) at a mobile phase flow rate of 0.8 mL/min and a temperature of 28 °C, through a stationary phase column that was 5 µm × 4.6 mm × 150 mm (Spherisorb ODS-1, Waters, Milford, MA, USA). Residual aflatoxins AFB_1_, AFB_2_, AFG_1_, and AFG_2_ were quantified at 363-nm excitation filter and 440-nm cut-off emission filter wavelengths using the fluorescence detector (RF-10AXL SMN C20954406285, Knauer, Berlin, Germany).

### 5.5. Estimation of Percent Aflatoxin-Binding Capacity

The aflatoxin-binding capacity of a material was determined by the percent of AFB_1_, AFB_2_, AFG_1_ or AFG_2_ adsorbed into it. Thus, the higher the aflatoxin-binding capacity, the lower the percent of unbound aflatoxin content in the buffer solution. The percent-binding capacity P_i_ of ith TBM/Mycobind^®^ in binding jth aflatoxin was determined using the model in Equation (2):

P_i_ = (IAT_ij_ − UAT_ij_)/IAT_ij_ × 100
(2)
where IAT_ij_ (ng/mL) is the initial concentration of jth aflatoxin in the test tube with ith TBM/Mycobind^®^; and UAT_ij_ (ng/mL) is the residual (unbound) jth aflatoxin in the test tube with ith TBM/Mycobind^®^ after the digestion period. The IAT_ij_ was considered to be the amount of aflatoxin recovered from the positive control adjusted by subtracting the value obtained for the negative control. The UAT_ij_ was adjusted by subtracting the residual aflatoxin amount that was obtained for the negative control of each individual TBM/Mycobind^®^ from the concentration of residual aflatoxin in the supernatant of the TBM/Mycobind^®^ spiked with aflatoxin solution.

### 5.6. Data Analysis

#### 5.6.1. Statistical Analyses

Data regarding the percent mean-binding capacity was analyzed by the General Linear Model (GLM) program of Statistical Analysis System (SAS) [41] using the model in Equation (3):Y_ij_ = X_i_ + X_j_ + e_ij_(3)
where
Y_ij_ = binding response (capacity) of the ith TBM/Mycobind^®^ in adsorbing the jth aflatoxinX_i_ = binding effect due to the capacity of the ith TBM/Mycobind^®^ in adsorbing the jth aflatoxin X_j_ = binding effect due to the ease with which the jth aflatoxin is adsorbed to the ith BMe_ij_ = the error term due to the ith and jth aflatoxin

The mean separation was done by the Duncan procedure, and the significance was declared at an alpha level of 0.05.

The relationship between the binding/adsorbing capacity of BM and the chemical composition of the BM was determined by running a correlation analysis between the percent binding/adsorbing capacity of BM and the chemical composition values (elemental (oxide) concentration and cation exchange capacity) of the BM. The correlation analysis of the data (Appendix A) was done by using MS Excel.

#### 5.6.2. Determination of Aflatoxin-Binding Capacity Ratio of Mycobind^®^ to the TBM

Binding capacity of Mycobind^®^ relative to a test binding material as a ratio *R* was determined using the model in Equation (4):
*R* = % binding capacity of Mycobind^®^ ÷ % binding capacity of a test binding material(4)

## Figures and Tables

**Table 1 toxins-10-00510-t001:** Percent of structural components in clay samples and rice husk ashes (RA) collected from various locations.

Samples (Number)	Mean/Range/One Value	Percent Structural Components of Clays and RA Samples					Source
		SiO_2_	Al_2_O_3_	Fe_2_O_3_	CaO	K_2_O	
Clays (11)	Mean	59.6	19.0	5.2	1.7	0.8	[21]
	Range	1.1–69.0	0.5–21.7	0.2–14.8	0.1–6.8	0.1–3.3	
Clays (10)	Mean	55.3	13.7	4.4	1.4	1.3	[22]
	Range	44.3–71.0	8.4–20.1	1.4–8.0	0.1–2.4	0.1–2.6	
Crude bentonite (1)	one value	80.2	13.2	2.7	0.2	0.1	[23]
RA (1)	one value	88.3	0.5	0.7	0.7	2.9	[24]
RA (1)	one value	89.0	1.2	1.3	1.0	1.2	[25]
RA (1)	one value	93.4	0.1	0.1	0.3	1.4	[26]

RA: Rice husk ash.

**Table 2 toxins-10-00510-t002:** Mineralogical and chemical composition of the test binding materials (TBM) and reference binder.

TBM and Mycobind^®^ ID	Prominent Minerals	Chemical Formula
AC	muscovite	KAl_2_(AlSi_3_O_10_)(F,OH)_2_
	hematite-proto	Fe_1.9_H_0.06_O_3_
KC	quartz	SiO_2_
	muscovite	KAl_2_(AlSi_3_O_10_)(F,OH)_2_
	lizardite	Mg_3_Si_2_O_5_(OH)_4_
CC	kaolinite	Al_2_Si_2_O_5_(OH)_4_
MC	kaolinite	Al_2_Si_2_O_5_(OH)_4_
	leucite	K[AlSi_2_O_6_]
	lizardite	Mg_3_Si_2_O_5_(OH)_4_
VA	pigeonite	(Ca, Mg, Fe) (Mg, Fe)Si_2_O_6_
	microcline	KAlSi_3_O_8_
	ephesite	NaLiAl_2_ (Al_2_Si_2_)O_10_(OH)_2_
RA	albite	NaAlSi_3_O_8_ or Na_1.0–0.9_Ca_0.0_
	terranovaite	NaCaAl_3_Si_17_O_40_8H_2_O
	sepiolite	Mg_4_Si_6_O_15_(OH)_2_·6H_2_O
R	metanatrolite	Na_2_Al_2_Si_3_O_10_
	phlogopite	KMg_3_(AlSi_3_O_10_)(F,OH)_2_
	andradite/melanite	Ca_3_Fe_2_(SiO_4_)_3_

In this and subsequent tables: clays from Arusha (AC), Kilimanjaro (KC), Coast (CC), and Morogoro (MC); volcanic ash (VA), rice husk ash (RA), and reference binder (R).

**Table 3 toxins-10-00510-t003:** The major elemental (oxide) composition of the test binding materials and reference binder.

BM	Elemental Oxide Composition of the BM (%)					CEC (meq/100 g)
	Al_2_O_3_	SiO_2_	K_2_O	CaO	Fe_2_O_3_	
AC	18.0	26.0	0.22	0.79	45.31	27.2
KC	25.0	31.0	0.01	0.24	39.73	18.8
CC	32.8	61.3	0.63	0.49	2.14	7.0
MC	24.0	34.8	0.52	0.54	36.1	15.4
VA	15.0	22.0	8.78	14.9	26.2	25.4
RA	0.5	75.7	9.54	1.71	0.59	27.2
R	5.1	49.0	6.99	13.4	19.8	38.9

In this and subsequent tables, BM = TBM and R, TBM = test binding materials, R = reference materials and CEC = cation exchange capacity.

**Table 4 toxins-10-00510-t004:** In vitro binding capacity of test binding materials and reference binder.

BM	Mean Percent of Bound Individual Aflatoxin				Mean Percent of Total Aflatoxin Bound	SEM
	AFB_1_	AFB_2_	AFG_1_	AFG_2_		
AC	97.9	60.6	99.9	32.2	72.6 ^a,b^	32.5
KC	95.4	40.1	96.1	14.5	61.5 ^b,c^	40.9
CC	96.6	14.4	31.3	17.3	39.9 ^c^	38.5
MC	95.6	32.6	94.6	25.3	62.0 ^b,c^	38.3
VA	97.9	28.9	71.5	30.7	57.3 ^b,c^	33.5
RA	94.6	79.8	91.5	72.7	84.7 ^a,b^	10.2
R	97.7	99.2	98.8	96.4	98.1 ^a^	1.3
Mean	96.5 ^a^	50.8 ^b^	83.4 ^a^	41.3 ^b^		
SEM	1.4	30.4	24.9	31.0		

SEM = standard error of the means; means with similar superscripts do not differ significantly.

**Table 5 toxins-10-00510-t005:** Relationship of the chemical composition of test binding material and reference binder with their aflatoxin-binding capacity.

BM		Elemental (Oxide) Composition of the BM (%)					CEC (meq/100 g)	MTAB (%)
		Al_2_O_3_	SiO_2_	K_2_O	CaO	Fe_2_O_3_		
AC		18.0	26.0	0.22	0.79	45.31	27.2	72.6
KC		25.0	31.0	0.01	0.24	39.73	18.8	61.5
CC		32.8	61.3	0.63	0.49	2.14	7.0	39.9
MC		24.0	34.8	0.52	0.54	36.1	15.4	62.0
VA		15.0	22.0	8.78	14.9	26.2	25.4	57.3
RA		0.5	75.7	9.54	1.71	0.59	27.2	84.7
R		5.1	49.0	6.99	13.4	19.8	38.9	98.1
Correlation coefficients	EC in BM and CEC	−086	−0.06	0.59	0.63	0.11		
	EC in BM and PBC	−0.88	0.21	0.51	0.34	0.00		
	CEC and PBC						0.90	0.90

BM = binding materials (TBM and R), TBM = test binding materials, R= reference binder, CEC = cation exchange capacity, EC = elemental (oxide) concentration, PBC = Percent binding capacity, MTAB = mean total aflatoxin-binding capacity of BM.

**Table 6 toxins-10-00510-t006:** Ratio of aflatoxin PBC of Mycobind^®^ to that of the test binding materials.

Aflatoxins	TBM					
	AC	KC	CC	MC	VA	RA
AFB_1_	1.0	1.0	1.0	1.0	1.0	1.0
AFG_1_	1.0	1.0	3.2	1.0	1.4	1.1
AFB_2_	1.6	2.5	6.9	3.0	3.4	1.2
AFG_2_	3.0	6.6	5.6	3.8	3.1	1.3
Average	1.0	2.0	3.0	2.0	2.0	1.0

PBC = Percent binding capacity, TBM = test binding materials.

**Table 7 toxins-10-00510-t007:** Physical appearance, sources, and current uses of the test materials.

Material ID	Physical Appearance	Source Region	Ethno-Utilization
**Clays**			
AC	Brick-red clogs	Arusha	Treatment of human skin infection and ailments
KC	Brownish-red blocks	Kilimanjaro	Geophagial satisfaction
CC	Shiny white granules	Coast	Stomach ailment treatment and for decorations
MC	Brownish-red granules	Morogoro	Geophagial satisfaction
**Ashes**			
VA	Greyish Volcanic powder	Arusha	Food seasoning and tenderization in traditional cookery, feed additive
RA	Greyish-white fine powder	Various places	Soil fertility improvement

In this and subsequent tables or figures: AC; Arusha clay, KC; Kilimanjaro clay, CC; Coast clay, MC; Morogoro clay, VA; volcanic ash, RA; rice husk ash.

**Table 8 toxins-10-00510-t008:** Experimental solutions.

Solution Samples	Composition	Number of Samples	Replications	Total # of Units (Tubes)
Spiked TBM/Mycobind^®^	Buffer solution, TBM/Mycobind^®^ and diluted aflatoxin solution	7	3	21
**Controls**				
Non-spiked TBM/Mycobind^®^	TBM/Mycobind^®^ and buffer solution	7	3	21
Positive control	Buffer solution and diluted aflatoxin solution	1	3	3
Negative control	Buffer solution	1	3	3

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
