# Peer review of "An In Vitro Evaluation of the Capacity of Local Tanzanian Crude Clay and Ash-Based Materials in Binding Aflatoxins in Solution"

_toxins, 2018, doi:10.3390/toxins10120510_

Reviewer 1 Report

The manuscript entitled "An in-vitro evaluation of the capacity of local Tanzanian crude clay and ash based materials in binding aflatoxins in solution" reports ned data on the use of local clay from Tanzania in binding aflatoxins in solution. The topic is intersting and adds information to the area of interest. The reduction  of feeds contamination by mycotoxins is a relevant issue and detoxification is also an open challenge. the manuscript fits the scope and aims of the Journal. It is properly written and results and conclusions are justified by data. It is suggested to add the following References in the text and in the Reference list:

Mikusova, P.; Ritieni, A.; Santini, A.; Juhasova, G.; Srobarova, A.  “Contamination by moulds of grape berries in Slovakia”.  Food Additives & Contaminants, Part A: Chemistry, Analysis, Control, Exposure & Risk Assessment  (2010),  27(5),  738-747.

Ketney O, Santini A, Oancea S. (2017). Recent aflatoxin survey data in milk and milk products: A review. International Journal of Dairy Technology, 70 (3), 1-12.

Santini A.and Ritieni A. “Aflatoxins: Risk, Exposure and Remediation” in Aflatoxins - Recent Advances and Future Prospects. Ed. Mehdi Razzaghi-Abyaneh, 2013, Chapter 16, 2013, ISBN: 978-953-51-0904-4, InTech Publisher, Rika,Croatia, (2013) 343-376.

Author Response

Seen the comments and find that the references given fit most in our two coming articles, one on in-vivo test of aflatoxin contamination of feeds and the other on the farmers' practice on use of contaminated foods and feeds.

Reviewer 2 Report

Review of “an in vitro evaluation of the capacity of local Tanzania and crude clay and ash based materials in binding aflatoxins in solution.”

 This paper contains novel and useful information and publication is recommended.  A number of relatively small concerns need to be addressed.

 Abstract

 Line 9. Mycobind.

 Line 10. The use of abbreviations for the materials tested means nothing to the reader and is not acceptable here. Either the names of these materials should be spelled out in the abstract or this and the following sentences condensed.

 Line 12 “TBM” needs to be defined.

 line 26. 27. Aflatoxin names require subscripts.

 Line 45, 46.  Simplify.  Suggest “A clear genetic linear relationship between binding effectiveness and absorbent properties, including elemental and mineralogical content, cation exchange capacity (CEC) and pH levels does not appear to exist (17, 18).”  Vekiru et al. 2007 does not need separate mention.

 Line 50 and elsewhere.  Elements and compounds should be written in all lower case.

 Line 54, 55.  The name of the imported binder should be inserted and this sentence requires a reference.

 Lines 57-59. This statement needs references or should be omitted.

 Results

 Lines 75-77.  All of these terms must be spelled out here and also in Table 2.

 Line 79.  As Table 5 in the text is mentioned next, it should be labelled Table 3, moved to the appropriate place and the succeeding tables renumbered.

 Line 104. “CEC” needs to be defined here and followed by “(CEC)”. Is not clear to this reader how the correlation coefficients were obtained or to what they pertain.

 Discussion

 Line 137. The meaning of “acidity promoter” is unclear.

 Line 144. What is an “X–RD” analysis”?

 Lines 154-173.  It seems logical that the description and discussion of the chemical composition values should be presented first in the Discussion section.

 Lines 199-201.  Perhaps geophagia has some influence on the effect of natural toxins on wild animals and birds but it seems very unlikely that an effect on aflatoxin has been described. These sentences should be omitted.

Author Response

The comment are really instrumental to improve the article. The following are responses for the comments. 

Reviewers’   comments

responses

Abstract

Line 9.   Mycobind.

Done

Line   10. The use of abbreviations for the materials tested means nothing to the   reader and is not acceptable here. Either the names of these materials should   be spelled out in the abstract or this or the following sentences condensed.

names   of these materials have been spelled out in the abstract

Line 12   “TBM” needs to be defined.

Done

Line   26. 27. Aflatoxin names require subscripts.

Done

Line   45, 46.  Simplify.  Suggest “A clear genetic linear relationship   between binding effectiveness and absorbent properties, including elemental   and mineralogical content, cation exchange capacity (CEC) and pH levels does   not appear to exist (17, 18).”  Vekiru et al. 2007 does not need   separate mention.

Corrected   and legibly written

Line 50   and elsewhere.  Elements and compounds should be written in all lower   case.

done

Line   54, 55.  The name of the imported binder should be inserted and this   sentence requires a reference.

An example   of imported binder was MycobindR a reference is personal communication   with one of the importers, that is, Farmers Centre Ltd

Lines   57-59. This statement needs references or should be omitted.

Reference   inserted

Results

Lines   75-77.  All of these terms must be spelled out here and also in Table 2.

Done

Line   79.  As Table 5 in the text is mentioned next, it should be labelled   Table 3, moved to the appropriate place and the succeeding tables renumbered.

Rearranged   and numbered sequentially

Line   104. “CEC” needs to be defined here and followed by “(CEC)”. Is not clear to   this reader how the correlation coefficients were obtained or to what they   pertain.

-CEC   defined

-Correlation coefficients   were determined by running correlation analysis between percent binding/adsorbing   capacity of BM and the chemical composition values (elemental-oxide   concentration and cation exchange capacity) of the BM using MS-Excel analysis   of the data (Supplementary Table S1)

Discussion

Line   137. The meaning of “acidity promoter” is unclear.

Written   legibly

Line   144. What is an “X–RD” analysis”?

X-RD is   defined X-ray diffraction

Lines   154-173.  It seems logical that the description and discussion of the   chemical composition values should be presented first in the Discussion   section.

Done accordingly

Lines   199-201.  Perhaps geophagia has some influence on the effect of natural   toxins on wild animals and birds but it seems very unlikely that an effect on   aflatoxin has been described. These sentences should be omitted.

-these animals and birds fetch and eat clayey soil which   render them safe from the inherent food toxins (Angel, 2007). Studies show   that clays   detoxify aflatoxins (Phillips et al., 2002, Denli et al., 2009 and Kaoud, 2013).